# Practitioners' perspectives on acupuncture treatment for postpartum depression: A qualitative study

Fan Liu[1,2]◉, Tian-yu Zhan[3]◉, Yu-qin Xu[1], Xiao-fei Lu[1], Yu-mei Zhou[1,2], Xing-xian Huang[1,2], Yuan-yuan Zhuo[1,2], Zhuo-xin Yang[1,2]*

1 The Fourth Clinical Medical College, Guangzhou University of Chinese Medicine, Shenzhen, China,
2 Department of Acupuncture and Moxibustion, Shenzhen Traditional Chinese Medicine Hospital, Shenzhen, China, 3 Community Health Service Management Centre, The Eighth Affiliated Hospital of Sun Yat-sen University, Shenzhen, China

◉ These authors contributed equally to this work.
* 001188@gzucm.edu.cn

**Data Availability Statement:** Data cannot be shared publicly because recordings of the conversations are private and contain potentially identifying and personal information. Data are

## Abstract

### Background

Acupuncture may become a treatment for postpartum depression (PPD). Currently, little is known about the use of acupuncture in the treatment of PPD from the point of view of practitioners. The aim of this study was to explore practitioners' perspectives on the treatment of PPD with acupuncture and provide suggestions for future improvement.

### Methods

This study employed a qualitative descriptive method. Semistructured, open-ended interviews were conducted with 14 acupuncture practitioners from 7 hospitals via face-to-face or telephone interviews. The data were collected using interview outline from March to May 2022 and analysed using qualitative content analysis.

### Results

In general, the use of acupuncture for treating PPD was positively regarded by practitioners. They claimed that acupuncture is both safe and helpful for breastfeeding women who are experiencing emotional discomfort and that it can alleviate a variety of somatic symptoms. The following three themes were extracted: (a) patient acceptance and compliance; (b) acupuncture as a treatment for PPD; and (c) the advantages and drawbacks of acupuncture treatment.

### Conclusion

Practitioners' optimistic outlooks demonstrated that acupuncture is a promising treatment option for PPD. However, the time cost was the most significant barrier to compliance. Future development will focus mostly on improving acupuncture equipment and the style of service.

available from the Ethics Committee of Shenzhen Traditional Chinese Medicine Hospital for researchers who meet the criteria for access to confidential data. The data underlying the results presented in the study are available via the following email address: szszyyll@126.com.

**Funding:** Fan Liu is supported by the Sanming Project of Medicine in Shenzhen (SZSM201612001).Zhuo-xin Yang is supported by National veteran Chinese medicine expert inheritance studio construction project ([2022]75). The funders had no role in study design, data collection and analysis, decision to publish, or preparation of the manuscript. There was no additional external funding received for this study.

**Competing interests:** The authors have declared that no competing interests exist.

# Introduction

Postpartum depression (PPD) is one of the most prevalent postpartum mental disorders and is indicated by low mood, a loss of interest, or exhaustion [1]. The global prevalence of PPD is estimated to be 17.22%. The PPD prevalence of 17.98% in mainland China is slightly higher than the global average [2]. It was predicted that the prevalence of PPD would increase even more during the COVID-19 pandemic [3, 4]. PPD not only affects maternal health but also adversely affects the mother-infant relationship and has a negative impact on the long-term physical and mental development of children [5, 6].

Effective treatments for PPD include psychosocial [7], psychological [7, 8], and pharmacological [9, 10] interventions, depending on the severity of the clinical presentation. Despite the availability of multiple treatment options, the remission rates in treatment studies vary considerably. Based on 17 studies examining remission rates for women with PPD who were treated with at least six weeks of an adequate antidepressant dose or evidence-based psychotherapy, the overall weighted mean remission rate of adequately treated women was 51.2% (95% confidence interval 48.6% to 53.8%) [11]. Therefore, a personalized medicine approach for PPD treatment has been advocated [12, 13]. Patient preferences are recognized as an integral part of personalized treatment. The majority of women with PPD prefer nonpharmacological therapies, especially while lactating, since they are concerned that drugs will harm their infant [14–16].

As a form of complementary and alternative medicine, acupuncture has been extensively utilized to treat psychiatric disorders. According to a cross-sectional survey, depression ranked second among the full spectrum of acupuncture indications, and acupuncture is a popular alternative for mental health management in the United States [17]. A nationwide survey in the United Kingdom revealed that psychological problems or symptoms were the second most common reason for patients to seek acupuncture [18].

Preliminary meta-analyses [19, 20] investigating the use of acupuncture for the management of PPD showed that acupuncture appeared to be effective with respect to certain outcomes, but further research is needed. A national multicentre registry study evaluating the effectiveness of different acupuncture treatments for PPD was recently conducted in China. In parallel, a qualitative exploration was undertaken to obtain the perspectives of acupuncture practitioners to assess the status quo of acupuncture for treating PPD and further examine the possible role of acupuncture as a supplementary therapeutic option for PPD. We also aimed to understand the potential barriers and facilitators to the implementation of acupuncture. This paper presents the findings from this qualitative study.

# Methods

## Study design

The study was performed from a constructivist point of view using a descriptive qualitative approach based on the general tenets of naturalistic inquiries [21, 22]. A naturalistic approach holds that realities need to be understood within a context, including the temporal context, and involves no prior commitment to any theoretical view of a target event. The aim of our study was to explore what acupuncture practitioners think about acupuncture for treating PPD. Little is known about this issue, so a qualitative interview approach was appropriate to offer a comprehensive summary or thorough description of the participants' understanding and attitudes. The design and reporting of our study were guided by the Consolidated Criteria for Reporting Qualitative Studies (COREQ) [23] and the Standards for Reporting Qualitative Research (SRQR) [24].

## Participants and setting

Semistructured, open-ended interviews were conducted to achieve the research aim. Acupuncturists from a multicentre registry study were invited to participate in the study. The registry study began in 2018 and was supported by the National Key Research and Development Plan of China (2017YFC1703604). The purpose of the study was to compare the effectiveness of different acupuncture treatments for PPD.

Based on a purposive sampling strategy, all frontline acupuncture practitioners in the registry study who met the following criteria were invited to participate in the qualitative interview study: practitioners who were involved in the clinical trial for more than 1 year and those who treated at least 20 patients. Ultimately, 16 acupuncturists from 7 Class-A tertiary hospitals met the requirements: One acupuncturist declined to participate, 1 did not respond, and 14 agreed to participate. The first author, who is based in Shenzhen, conducted all interviews face-to-face or by telephone, depending on the participants' locations. The interviewer is male, with a Ph. D. in Chinese medicine; he was a postdoc at the time of the study and has had training and experience in qualitative research since 2019.

The acupuncture practitioners' demographic information is presented in Table 1. The characteristics of the interviewees are described, including their sex, age, educational background, professional title, years of experience and region.

A prior appointment was set with each participant to ensure that the interview would proceed without interruption or distraction. No one else was present during the face-to-face interviews, which were held in each doctor's office. The researcher was responsible for monitoring the interview process and ensuring that all topics on the agenda were discussed within the allotted time.

**Table 1. Demographic information of the participants.**

|  |  | N = 14 |
|---|---|---|
| Sex (n, %) | Female | 13(92.9) |
|  | Male | 1(7.1) |
| Age (n, %) | <30 years | 3(21.4) |
|  | 30–40 years | 5(35.7) |
|  | 40–50 years | 4(28.6) |
|  | 50–60 years | 2(14.3) |
| Educational background (n, %) | Master's | 10(71.4) |
|  | Doctorate | 4(28.6) |
| Professional title (n, %) | Resident physician | 3(21.4) |
|  | Attending physician | 4(28.6) |
|  | Associate chief physician | 3(21.4) |
|  | Chief physician | 4(28.6) |
| Years of experience (n, %) | <5 years | 3(21.4) |
|  | 5–10 years | 4(28.6) |
|  | 10–20 years | 4(28.6) |
|  | 20–30 years | 3(21.4) |
| Region (n, %) | South China | 11 (78.6) |
|  | North China | 3(21.4) |
| Type of interview (n, %) | Face-to-face | 9 (64.3) |
|  | Telephone | 5(35.7) |
| Relationship to the interviewer (n, %) | Knew before the interview | 10(71.4) |
|  | First met at the interview | 4(28.6) |

## Ethics

The study protocol was approved by the Ethics Committee of Shenzhen Traditional Chinese Medicine Hospital (Approval no. K2022-008-01) on February 28,2022. Written informed consent was obtained from all participants for inclusion in the study.

## Data collection

The main line of the interview outline revolved around the process of the registry study (Fig 1). The question schedule was formulated by the research team (FL, TYZ, YMZ, YYZ and ZXY) and included 13 questions within 3 subjects (S2 Table).

The data were collected by the researcher from March to May 2022 through one-on-one interviews. All 14 participants completed two interviews. Informed consent was obtained from every subject at the beginning of the interview. Demographic data were collected near the end of the interview.

During the first interview, the interview outline was used to provide a semistructured approach and allow flexibility to uncover topics presented by the participants that were not

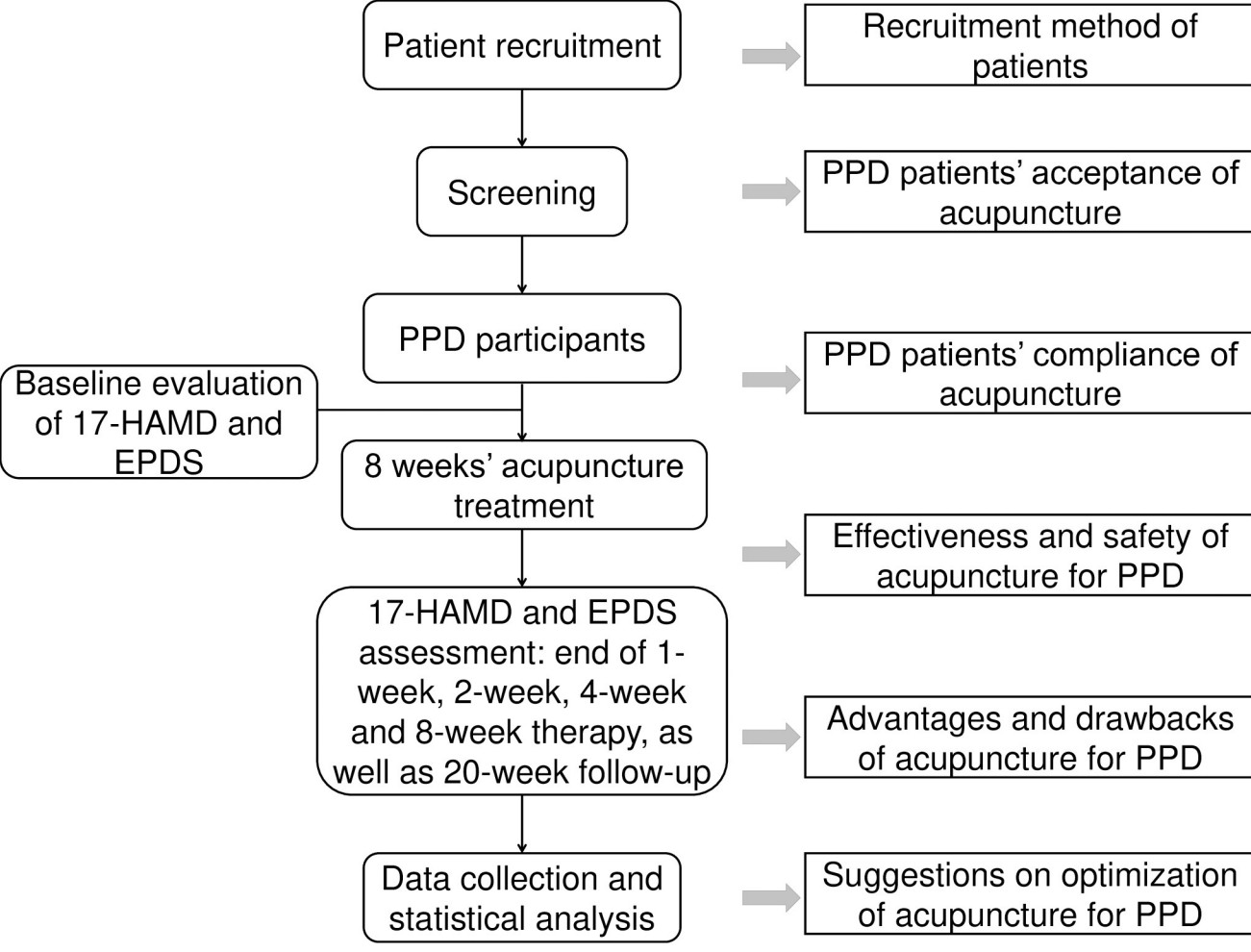

**Fig 1. Flowchart of the registry study design and the questions for each process.** EPDS, Edinburgh Postnatal Depression Scale; 17-HAMD, 17-item Hamilton Rating Scale.

included in the question schedule. A second interview was conducted to supplement and confirm the content of the first interview.

The interview sessions averaged approximately 32 minutes for each participant and were audio-recorded. The interview started with the introduction of the purpose of the research and the interview process. Then, the interviewee was asked about his or her working experience in the registry study, followed by the open-ended questions included in the interview outline. Before concluding the interview, the interviewer summarized the topics to avoid missing details and expressed gratitude to the interviewees for their valuable time.

## Data analysis

The interview recordings were transcribed using iFlytek Sr501 smart voice recorder; then, YQX and XFL checked the transcripts and submitted the final transcripts to the interviewees for confirmation. After ensuring that the transcripts were correct, they were deidentified and imported into Excel for qualitative analysis. The transcripts were analysed using content qualitative analysis [25, 26]. The researchers (FL and TYZ) began data analysis before completing data collection. They immersed themselves in the data by reading the transcripts multiple times. A preliminary code list, including possible themes and specific codes, was developed by the research team (FL, TYZ, YQX, and XFL) following discussion. FL and TYZ coded the data independently. Any differences arising from the whole work were resolved by consensus. YQX and XFL participated in finalizing the code structure. All analyses were performed in Chinese and then translated into English for reporting.

## Results

### Overview

In general, the use of acupuncture for treating PPD was positively regarded by the practitioners. They claimed that acupuncture is both safe and helpful for breastfeeding women who are experiencing emotional discomfort and that it can alleviate a variety of somatic symptoms. Patient acceptance of acupuncture was rather high because of its lengthy history and widespread use in China. However, patient compliance was hindered by the high time cost. The acupuncture practitioners offered several suggestions for improvement to better assist individuals suffering from PPD. The results are presented according to the final categorical framework, including (a) patient acceptance and compliance, (b) acupuncture as a treatment for PPD, and (c) the advantages and drawbacks of acupuncture treatment. Examples of meaning units, condensed meaning units, codes, subthemes and themes from the content analysis are presented in Table 2.

**Table 2. Qualitative analysis process.**

| Meaning unit | Condensed meaning unit | Code | Subtheme | Theme |
|---|---|---|---|---|
| *I think that 80% of them (the patients) are open to acupuncture. Acupuncture is recognized by the public rather than treated as a novel treatment. (Practitioner 02, Resident physician)* | *80% of patients are open to acupuncture.* | High acceptance | Acceptance | Patient acceptance and compliance |
| *Acceptance is also influenced by the qualification of the therapist. The patient may be concerned if the therapist is a novice or young. (Practitioner 02, Resident physician)* | *Acceptance is influenced by the qualification of the therapist.* | Influence factor | | |
| *Establishing a harmonious and friendly doctor–patient relationship is more conducive to improving patient compliance. (Practitioner 05, Chief physician)* | *Good doctor–patient relationship helps to improve compliance.* | Influence factor | Compliance | |

*(Continued)*

**Table 2.** (Continued)

| Meaning unit | Condensed meaning unit | Code | Subtheme | Theme |
|---|---|---|---|---|
| *It's hard for mothers to take care of their babies for a long time. They have to hold their babies frequently for breastfeeding. Therefore, waist and back pain are very common. (Practitioner 06, Associate chief physician)* | *Waist and back pain are very common.* | Pain problem | Chief complaints and symptoms | Acupuncture as a treatment for PPD |
| *Many mothers seek treatment for sleep-related issues, with the primary goal of resolving the problem. Because a good night's sleep leads to a good mood, the two are linked. (Practitioner 14, Attending physician)* | *Many mothers seek treatment for sleep-related issues.* | Sleep problem | | |
| *I base my practice on the idea of person-centred care. When the mother arrives, for example, she might be tired from holding the infant and have wrist, neck, or low back pain. I would occasionally do Qihuang needle therapy for her after this treatment plan (Tiaoren Tongdu), and the pain would be relieved quickly. (Practitioner 06, Associate chief physician)* | *I base my practice on the idea of person-centred care.* | Treatment concept | Acupuncture treatment methods | |
| *Acupuncture was of great help to the body. For example, patients reported being able to sleep, eating well, having less physical discomfort, and being more interested in other things. The whole essence, Qi and spirit were better. (Practitioner 08, Chief physician)* | *Acupuncture was of great help to the body.* | Positive comment | Effectiveness and safety | |
| *From my personal clinical point of view, the frequency of treatment affects the therapeutic impact. Two to three times each week is guaranteed to be more effective. (Practitioner 05, Chief physician)* | *The frequency of treatment affects the therapeutic impact.* | Influence factor | | |
| *Both the symptoms and the root cause need to be treated. If the patient was worried about the pain, we could help her address it jointly, making her psychological condition much better. . . .We are traditional Chinese medicine practitioners. We consider patient care from the perspective of patients and put patients first to alleviate their discomfort. (Practitioner 06, Associate chief physician)* | *To treat both symptoms and the root cause from a holistic view.* | Holistic view | Advantages | Advantages and drawbacks of acupuncture treatment |
| *Acupuncture treatments are reasonably inexpensive. Furthermore, medical insurance can be used, and the financial burden on patients is minimal. (Practitioner 09, Attending physician)* | *The cost of acupuncture treatment is relatively low.* | Treatment cost | | |
| *The entire treatment cycle was rather long, and living so far away was inconvenient for the patients. If a patient lived far away, the drive may take two hours plus the time for acupuncture, for a total of three or four hours, during which time she might have breast swelling. (Practitioner 06, Associate chief physician)* | *Acupuncture treatment takes more time.* | Time cost | Drawbacks | |
| *Acupuncture is applicable for mild to moderate PPD. If patients have suicidal thoughts, there is still a risk when using acupuncture alone. (Practitioner 07, Chief physician)* | *Acupuncture is applicable for mild to moderate PPD.* | Scope of application | | |
| *We can use acupoint catgut embedding appropriately and embed the catgut at the main points according to the scheme of Tiaoren Tongdu so that the patients don't need to come so often. . . Acupoint catgut embedding is suitable for many chronic diseases. It has good effects on chronic diseases of the digestive system and respiratory system and can prolong the efficacy of acupuncture. (Practitioner 06, Associate chief physician)* | *Acupoint catgut embedding is recommended for use.* | Acupuncture equipment | Suggestions | |
| *I believed that door-to-door service was beneficial for PPD patients who were still in confinement, since rushing between their homes and hospitals would increase numerous risk factors, such as anxiety and getting a cold, with potentially negative consequences. (Practitioner 09, Attending physician)* | *Door-to-door service was beneficial for PPD patients.* | Door-to-door service | | |
| *Because the disease is PPD, we should refer to the psychological clinic's model to enhance a patient's experience of visiting a doctor and strive to offer a private room with a pleasant environment. (Practitioner 01, Attending physician)* | *We should refer to the psychological clinic's model to enhance a patient experience.* | Hospital environment | | |

## Theme 1: Patient acceptance and compliance

Acupuncture is widely accepted by the general public in China as an essential nondrug therapy in traditional Chinese medicine. The practitioners claimed that patients with PPD might be surprised to learn that acupuncture can help with PPD but that the majority of them were willing to try it.

*I think that 80% of them (the patients) are open to acupuncture. Acupuncture is recognized by the public rather than treated as a novel treatment. Only a few patients explicitly refused because they were afraid of the pain of acupuncture. (Practitioner 02, Resident physician)*

*In fact, many people ask whether acupuncture can treat depression because most individuals do not make a connection between the two. I generally tell her (the patient) that we care about her as a whole person, not just about her depression. We require a complete picture of you. Acupuncture can help balance your body's yin and yang and alleviate a variety of illnesses. For example, you stated that you have back pain and sleep issues. Because your body and psyche are one, improving the balance between yin and yang would enhance your whole state, physical symptoms, and mood. That was my explanation to them. (Practitioner 14, Attending physician)*

Prior acupuncture experience, the hospital's reputation, the qualification of the acupuncturists, and the presence or absence of physical symptoms were the key factors determining the patients' acceptance. Patients who had a previous pleasant acupuncture experience for other health problems had a higher level of acupuncture acceptance.

*If a patient had previously received acupuncture therapy and felt that it was useful, whether for cervical and lumbar spondylosis or specifically for postpartum pelvic floor muscle rehabilitation, they were more willing to accept it. (Practitioner 03, Resident physician)*

Patients were more inclined to believe in the benefits of and accept acupuncture treatment from high-level facilities and senior doctors.

*First, our department is a national key clinical specialty in acupuncture, and it should be No. 1 in Shenzhen. Second, we have numerous associate chief physicians and chief physicians who have worked for more than 10 years. As is customary, our clinic is quite busy, and patients are occasionally unable to register. They prefer to adopt it (acupuncture) because we are confident in our skills. (Practitioner 06, Associate chief physician)*

*Acceptance is also influenced by the qualification of the therapist. A patient may be concerned if their therapist is a novice or young. (Practitioner 02, Resident physician)*

Furthermore, patients with apparent physical symptoms, such as pain or insomnia, were more receptive to acupuncture treatment.

*Patients with physical discomfort are more likely to receive acupuncture because if she (the patient) is only slightly emotionally depressed and has no physical problems, her desire for treatment is low, and she might believe she can make adjustments by herself. Alternatively, she might think her discomfort is caused by a problem with her husband or mother-in-law. She would be alright as long as they (her husband and mother-in-law) were okay, so she would not need acupuncture at all. However, patients with physical issues might believe that*

*acupuncture will improve their health anyway. For this reason alone, they might be willing to try it. (Practitioner 14, Attending physician)*

*If a patient is experiencing sleeplessness, anxiety, and digestive system issues, and felt uneasy everywhere, she might be unsure which department to visit for these symptoms. She might believe that if she could be treated with acupuncture, there would be hope. (Practitioner 09, Attending physician)*

Fear of pain or discomfort was the primary reason patients declined acupuncture treatment. The majority of these patients had never experienced acupuncture previously, and their irrational dread amplified the imaginary pain.

*Many people are afraid of acupuncture, especially the feeling of acupuncture. A person who has never been exposed to acupuncture would instinctively reject this invasive operation. The discomfort of acupuncture may be connected to the tension it causes for many people. When the needle punctures her skin, the more tense she (the patient) is, the tighter her muscles will be, and the stronger she will feel it (the puncture). However, if you let her try to become calm, she might not be able to relax on her own, so I will let her take a deep breath and advise her to breathe in and out gently. Next time, she will be less stressed and have less unpleasant feelings. (Practitioner 08, Chief physician)*

Treatment compliance with acupuncture for PPD needs to be improved. Data from the registry study's interim analysis revealed that more than 40% of the patients did not complete the course of treatment (8 weeks, 24 times) for various reasons.

The time cost was the most important factor affecting patient compliance. Long commutes due to poor transportation or vast distances could make it difficult to adhere to treatment. The pandemic also made it inconvenient for patients to come to the hospital for treatment. If a patient had returned to work after maternity leave, it would certainly have been a challenge to continue treatment. Moreover, family members' support was also important because the postpartum mothers were under great pressure with breastfeeding and childcare, and it was hard to make time for regular treatment without help from their family. The baby's health status was also crucial. If her child was frequently ill, the mother expended more effort caring for her child, reducing the time available for her own treatment.

*The lack of time is the most significant problem affecting patients' therapy. Some live far away, and they(patients) may feel that it is too troublesome. Living so far away requires too much time on the road. Another reason why a patient may have no time is that she has to take care of her children. In fact, many of our postpartum mothers took care of their children alone, and they brought the babies to the hospital during treatment. Some women claimed they had already started working. They felt too busy to receive treatment because they had to return home after work to see their children. Time, distance, and the pandemic scenario are the key causes. (Practitioner 11, Associate chief physician)*

*Some patients live far away, making frequent visits to the hospital inconvenient. Some mothers may have no spare time for treatment after taking care of their sick babies. Anyway, I think factors related to the child have a greater impact on treatment compliance. In truth, patients were aware of the problem and wished to undergo treatment, but their circumstances made this impossible. Furthermore, owing to the pandemic scenario, some people were unable to arrive since they were restricted or quarantined. (Practitioner 06, Associate chief physician)*

*Time is the most important factor, as patients need to breastfeed (and take care of their children). In fact, there were a few patients who brought their children, which is a great obstacle. If the family were more supportive, it would be better to have the father take care of the children and send the patients over. However, the mothers now arrive alone and are rarely accompanied by their relatives. Some moms bring their child, even two children, so that the older child may look after the little one during the treatment. Therefore, I think time is a great problem. (Practitioner 05, Chief physician)*

Anxious patients were reported to be less likely to comply with treatment. Some patients might be in a hurry to eliminate their discomfort, and if they did not see an improvement after several treatments, they were likely to abandon acupuncture treatment.

*I was also impressed by another patient. I thought acupuncture didn't help her. The patient was in a hurry when she came and had severe insomnia. She said that she hadn't had a good night's sleep for many days, and she had never felt like this before, which made her very anxious. She just came twice and felt no effect, so she didn't come anymore. She was too nervous and hoped that after one session of acupuncture, she could go back to sleep at night. (Practitioner 04, Resident physician)*

Of course, many practitioners mentioned positive benefits, and that a favourable doctor–patient relationship enhanced adherence. If a patient had enough trust in their doctor, they would generally follow the doctor's advice about the treatment schedule.

*Establishing a harmonious and friendly doctor–patient relationship is more conducive to improving patient compliance. (Practitioner 05, Chief physician)*

*Communication between the doctor and the patient is also crucial. Patients will initiate conversations with their doctors, and as a result, they will have greater faith in their therapists or therapy, which helps to improve compliance. (Practitioner 01, Attending physician)*

### Theme 2: Acupuncture as a treatment for PPD

**Chief complaints and symptoms.**   Practitioners stated that the most common complaints of patients with PPD focused on their emotions, sleep, and pain. The main symptoms of depression included sadness, a loss of interest, irritability, self-blame, and guilt.

Breastfeeding at night has an impact on sleep, causing issues such as difficulty falling asleep, light sleep, dreaminess, and easy awakening.

*Many patients had problems with poor sleep. There were two kinds of poor sleep. The first was because the patient couldn't fall asleep on her own, and the second was because she had to wake up passively (at night) to breastfeed. The latter kind was more common. (Practitioner 12, Attending physician)*

*Many mothers seek treatment for sleep-related issues, with the primary goal of resolving the problem. Because a good night's sleep leads to a good mood, the two are linked. (Practitioner 14, Attending physician)*

Pain was another typical symptom reported by patients. The locations that caused the most discomfort were the waist, back, shoulders, neck, and wrists.

*It's hard for mothers to take care of their babies for a long time. They have to hold their babies frequently for breastfeeding. Therefore, waist and back pain are very common. (Practitioner 06, Associate chief physician)*

Furthermore, PPD was frequently accompanied by digestive symptoms such as abdominal distension, pain, constipation, and a loss of appetite. Finally, patients often reported symptoms of memory loss, weariness, urinary incontinence, sweating, and palpitation.

**Acupuncture treatment methods.** The clinical study recommends filiform needle acupuncture with acupoint selection according to Tiaoren Tongdu (points: GV20, GV29, CV12, CV6, CV4, PC6, HT7, LI4, ST36, SP6, LR3). In this study, the majority of participants received only filiform needle acupuncture. But since this study was a real world study, different acupuncture practitioners adopted different acupuncture techniques, such as electroacupuncture and auricular acupuncture, based on their academic training and clinical experience. Additionally, combination therapies, such as cupping and moxibustion, could be used as needed. Acupuncturists were permitted to add or remove acupoints and select the appropriate combination therapy based on the actual condition of a patient.

*I base my practice on the idea of person-centred care. When a mother arrives, for example, she might be tired from holding the infant and have wrist, neck, or low back pain. I would occasionally do Qihuang needle therapy for her after this treatment plan (Tiaoren Tongdu), and the pain would be relieved quickly. (Practitioner 06, Associate chief physician)*

*For patients with low back muscle soreness, for example, I use cupping treatment to relieve symptoms rapidly after acupuncture. (Practitioner 10, Chief physician)*

**Effectiveness and safety.** Generally, the acupuncturists were satisfied with the performance of acupuncture in treating PPD, particularly physical symptoms. They said that acupuncture had a fantastic immediate effect that could help patients with their discomfort issues. In addition, they were optimistic about the long-term curative impact and expected acupuncture to reduce the recurrence rate.

*I thought acupuncture treatment could not only give patients psychological support through the communication between the physicians and their patients but also solve their physical discomfort. Some patients, for example, said that after acupuncture, they could fall asleep easily or that their backache was relieved instantly. . . Acupuncture might pay greater attention to relieving bodily issues while also improving overall vitality. (Practitioner 03, Resident physician)*

*Acupuncture was of great help to the body. For example, patients reported being able to sleep, eating well, having less physical discomfort, and being more interested in other things. The whole essence, Qi and spirit were better. (Practitioner 08, Chief physician)*

*Acupuncture was useful in treating somatic symptoms, including gastrointestinal problems. There would usually be feedback after two or three attempts. Some patients even sent me messages immediately after treatment, claiming that their symptoms had improved and that they were no longer as uncomfortable. Their symptoms might still be present, but they did not seem to be that severe. (Practitioner 11, Associate chief physician)*

*Among the patients I treated, the recurrence rate of depression was relatively low after two months of acupuncture treatment. (Practitioner 07, Chief physician)*

The acupuncturists noted that the therapy effectiveness was proportional to the treatment frequency. Acupuncture therapy 2–3 times per week was usually sufficient to ensure a curative effect.

*From my personal clinical point of view, the frequency of treatment affects the therapeutic impact. Two to three times each week is guaranteed to be more effective. Not only patients with PPD but also other outpatients are at risk of recurrence if they are only treated once a week. (Practitioner 05, Chief physician)*

The family relationship should also be examined. The effect of acupuncture treatment was considerably diminished if the patients with PPD had tense relationships with their mothers-in-law, had experienced infidelity, or had children with congenital disorders.

*If a patient's family conflicts had not been resolved, the therapeutic effect of acupuncture would have been limited in improving her sleep and physical pain at the time rather than addressing the root cause of her depression. She might continue to have depression as long as she has family issues. (Practitioner 02, Resident physician)*

Acupuncturists' humanistic care was crucial to increasing patient trust. Patients with greater trust in acupuncture and their therapists had better treatment outcomes, according to some acupuncturists.

*One case that impressed me deeply was a patient who was a well-educated and middle-class woman who worked in a public welfare job for a nonprofit organization. She struck me as a compassionate and kind person. When I performed a psychological assessment of her for the first time, she suddenly broke into tears. I asked her how she felt now. She said that it was because, for the first time, someone was so meticulous in sorting out her mood during this period. She felt very valued. She had a strong feeling of being seen and understood, and I believe the treatment started working at that point. (Practitioner 04, Resident physician)*

*Patients who are very active in their acupuncture treatment tend to have better treatment effects. On the one hand, I think it's the function of acupuncture itself. On the other hand, I communicate with my patients, which allows the patient to vent their emotions. They come to talk about their moods regularly, a few days a week. (Practitioner 05, Chief physician)*

Acupuncture is well known for its safety. Adverse reactions such as needle fainting, subcutaneous bleeding, and bruises occur occasionally. Patients should be informed prior to therapy.

*I don't think there is any problem with the safety of acupuncture. The most common adverse event recorded here is bruising because acupuncture is an invasive operation, which will inevitably damage some small capillaries, and the bruising will be absorbed slowly. Generally, patients will accept this if it is discussed with them in advance. (Practitioner 05, Chief physician)*

## Theme 3: Advantages and drawbacks of acupuncture treatment

**Advantages.** According to the acupuncturists, the major advantage of acupuncture in the treatment of PPD is that it is harmless and nontoxic, making it ideal for breastfeeding mothers.

The theoretical foundation of acupuncture is the unification of body and mind, as well as holistic balance. Individualized and patient-centred treatment can address a variety of symptoms in PPD patients while also allowing treatment regimens to be flexible.

*Both the symptoms and the root cause need to be treated. If a patient is worried about the pain, we can help her address it jointly, making her psychological condition much better...We are traditional Chinese medicine practitioners. We consider patient care from the perspective of patients and put patients first to alleviate their discomfort. (Practitioner 06, Associate chief physician)*

*Overall conditioning can be achieved using acupuncture. It is based on related symptoms, similar to traditional Chinese medicine. I won't tell a patient that they don't sleep well and that I'll treat them for it; if they're sad, I'll treat them for it. If it is accompanied by additional discomfort, as I previously stated, we will solve it together. (Practitioner 08, Chief physician)*

*A patient of mine had a fantastic metaphor. She said that she also had depression a long time ago, before she had a child. She took drugs and got better. She felt like she was soaking in warm water. She got warm because of the water temperature. However, this time, when she came for acupuncture treatment, she noticed that she was warming up from the inside out. She could mobilize her internal strength to a greater extent to help her through such a period. (Practitioner 04, Resident physician)*

Acupuncture is more accessible than psychotherapy in China because it is commonly available in hospitals at all levels, be it Chinese medicine hospitals or general hospitals. The price of acupuncture is acceptable for the general public.

*Acupuncture treatments are reasonably inexpensive. Furthermore, medical insurance can be used, and the financial burden on patients is minimal. (Practitioner 09, Attending physician)*

**Drawbacks.** The deficiency of acupuncture treatment mainly lies in the high time cost. This limits the widespread application of acupuncture in patients with PPD.

*The entire treatment cycle was rather long, and living so far away was inconvenient for the patients. If a patient lived far away, the drive may take two hours plus the time for acupuncture, for a total of three or four hours, during which time she might have breast swelling. (Practitioner 06, Associate chief physician)*

Another drawback is that acupuncture cannot be used alone to treat severe PPD patients with suicidal thoughts; it must be used in conjunction with other treatments, such as medications.

*Acupuncture is applicable for mild to moderate PPD. If patients have suicidal thoughts, there is still a risk when using acupuncture alone. (Practitioner 07, Chief physician)*

**Suggestions.** The acupuncturists recommended numerous improvement ideas to better alleviate PPD. They advised that the amount of stimulation be increased, utilizing methods such as acupoint burying and needle pressing, to retain efficacy in response to the inconvenient conditions that patients frequently present with at the hospital.

*We can use acupoint catgut embedding appropriately and embed the catgut at the main points according to the scheme of Tiaoren Tongdu so that the patients don't need to come so often. . . Acupoint catgut embedding is suitable for many chronic diseases. It has good effects on chronic diseases of the digestive system and respiratory system and can prolong the efficacy of acupuncture. (Practitioner 06, Associate chief physician)*

*Because of the wide variety of acupuncture treatments, multiple acupuncture therapies can be combined to accomplish the goal of synergistic enhancement. (Practitioner 13, Associate chief physician)*

In addition, the therapists proposed that acupuncture be more widely available in community health service centres and that regular therapy be provided through a door-to-door service by family doctors.

*I believed that the door-to-door service was beneficial for PPD patients who were still in confinement, since rushing between their homes and hospitals would increase numerous risk factors, such as anxiety and getting a cold, with potentially negative consequences. (Practitioner 09, Attending physician)*

The upgrading of the treatment environment could also contribute to the enhancement of the treatment experience, which is crucial for the PPD patient population. According to some therapists, thinner acupuncture needles and trocars can be applied to minimize the pain caused by needle insertion.

*Because the disease is PPD, we should refer to the psychological clinic's model to enhance a patient's experience of visiting a doctor and strive to offer a private room with a pleasant environment. (Practitioner 01, Attending physician)*

*I suppose we need to improve the entire treatment experience, making patients feel relaxed and valued. The treatment environment is critical (to the curative effect), including noise, privacy, and the time spent waiting. In public hospitals, conditions may be constrained. (Practitioner 02, Resident physician)*

## Discussion

Acupuncturists' perspectives on the present state of acupuncture in the treatment of PPD were explored in this qualitative study. We learned about the possible barriers and facilitators to the acceptability, compliance, and effectiveness of acupuncture; the benefits and drawbacks of acupuncture in the treatment of PPD; and suggestions on how to improve acupuncture for treating PPD.

The main advantages of acupuncture for treating PPD, according to the findings of this study, are its safety, effectiveness, accessibility, and affordability. Compared with drug therapy, patients are more likely to accept acupuncture treatment and other nondrug therapies. Compared with psychotherapy, acupuncture has a more direct effect on improving somatic symptoms and is more accessible and affordable in China.

Although high-quality evidence of acupuncture for depression is currently insufficient [27], practitioners have reported that it has a good effect on PPD. Given the safety of acupuncture [28], it has the potential to be a therapeutic option for PPD. Acupuncture was considered to have beneficial therapeutic outcomes on sleep and pain problems related to PPD. Another

qualitative study [29] showed that acupuncturists were more focused on physical symptoms and whether these symptoms could be resolved by acupuncture to speed up the improvements in the symptoms of depression. Meta-analyses [30, 31] have shown the effectiveness of acupuncture in the field of pain management, and related molecular mechanisms have been revealed [32]. A recent RCT [33] found that electroacupuncture may improve the sleep quality of patients with depression. A meta-analysis of RCTs [34] supported the use of acupuncture as an effective treatment to improve symptoms of depression-related insomnia.

However, acupuncture for PPD still has limitations that need to be addressed. First, according to this study, the majority of patients are unaware that acupuncture can help with PPD. Acupuncture is more commonly used to treat neurological problems in China than in Western countries. Stroke is one of the few main ailments for which the Chinese population believes acupuncture is the best treatment [35, 36]. Many respondents indicated that the promotion of acupuncture treatment for depression and other emotional issues should be intensified. Second, a lack of time in patients with PPD is a very common problem due to childcare issues. This study showed that the time cost was the most important issue affecting patient compliance. Consistent with the findings of a questionnaire study [15], a lack of time was the greatest perceived potential barrier to treatment for perinatal depression. Many patients did not have the time to commit to three sessions each week. Moreover, the respondents believe that the frequency of treatment is critical to its effectiveness. According to a recent systematic review and meta-analysis of acupuncture for depression [37], "both the total number of treatments and the frequency of treatments may play a role in depression-related outcomes." In addition, a review [38] studied the possible reasons for the negative results of international clinical trials on acupuncture and found that treatment frequency was too low in most studies, and the studies with positive results had a significantly higher treatment frequency.

In view of this problem, some practitioners advocate for improved acupuncture equipment, such as catgut thread and intradermal thumbtack needles, which can be retained at acupoints for longer periods of time to increase the amount of simulation to sustain the curative effect. Clinically, acupoint catgut embedding therapy is generally used for the treatment of obesity [39], and there is still a lack of research on this therapy for the treatment of depression. Intradermal thumbtack needles have emerged as a new type of embedding therapy with the evolution of acupuncture apparatuses in recent years. It is a shallow needling technique that reduces pain and extends the acupuncture effect by keeping the needle in place for a longer period of time. To increase stimulation, patients can self-press the needle. However, related clinical research is still lacking and should be carried out in the future.

Another prominent suggestion from practitioners was to optimize the acupuncture service model and provide door-to-door acupuncture treatment for patients with PPD. Despite the fact that primary health care in China continues to confront numerous challenges [40, 41], the popularization and advancement of community-based primary care networks in recent years is evident to all. Shenzhen, China's first Special Economic Zone, paved the way in building a primary care system with comprehensive community health care centres. Postpartum home visits have become a component of community health care services. A qualitative study [42] of early postpartum mothers in Shenzhen showed that home visits contributed to the new mothers' psychological well-being. Postpartum visits should focus on the mental health of mothers and provide further support. According to the Shenzhen community health service institution setting standards, community health care centres must be equipped with a certain percentage of Chinese medicine doctors. Policymakers should consider home-based acupuncture services to improve the convenience of acupuncture for patients with PPD.

## Limitations

Several limitations of the present study should be noted. Because our qualitative interview study was nested within a registry study, the sample we could choose from was limited, and information saturation could not be guaranteed. It is possible that the findings of this study may be nonrepresentative since acupuncture is practised in many different ways in China. Our respondents practised in public hospitals, and a broader perspective requires the participation of acupuncture practitioners from different regions and backgrounds. In addition, acupuncturists are not neutral in their assessment of the effects of acupuncture on PPD, which may lead to overly optimistic assessments.

## Conclusion

To the best of our knowledge, this qualitative exploration is the first to investigate the opinions of practitioners regarding the current situation, problems, and suggestions of acupuncture treatment for PPD. Practitioners' positive attitudes suggest that acupuncture is a promising treatment for PPD because it is a safe, nondrug therapy that is suitable for the postnatal period. One of the major advantages of acupuncture treatment is the ability to give individualized holistic treatment according to different conditions and symptoms. However, time consumption is the main barrier that hinders the implementation of acupuncture. Future directions mainly include improving acupuncture equipment and the service mode.

## Supporting information

**S1 Table. Consolidated criteria for reporting qualitative research (COREQ): A 32-item checklist for interviews and focus groups.**
(DOCX)

**S2 Table. The outline of the interview.**
(DOCX)

## Acknowledgments

The authors would like to recognize and thank all the participants for voluntarily participating in this study.

## Author Contributions

**Conceptualization:** Fan Liu, Tian-yu Zhan, Zhuo-xin Yang.

**Data curation:** Fan Liu, Tian-yu Zhan, Yu-qin Xu, Xiao-fei Lu.

**Formal analysis:** Fan Liu, Tian-yu Zhan.

**Investigation:** Fan Liu.

**Methodology:** Fan Liu.

**Supervision:** Yu-mei Zhou, Xing-xian Huang, Yuan-yuan Zhuo, Zhuo-xin Yang.

**Writing – original draft:** Fan Liu.

**Writing – review & editing:** Fan Liu, Tian-yu Zhan, Yuan-yuan Zhuo, Zhuo-xin Yang.

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
