## [Decision Letter · Decision Letter 0]

11 Aug 2022

PONE-D-22-18861Practitioners’ perspectives on acupuncture treatment for postpartum depression: A qualitative studyPLOS ONE

Dear Dr. Zhuoxin Yang,

Thank you for submitting your manuscript to PLOS ONE. After careful consideration, we feel that it has merit but does not fully meet PLOS ONE’s publication criteria as it currently stands. Therefore, we invite you to submit a revised version of the manuscript that addresses the points raised during the review process.

We look forward to receiving your revised manuscript.

Kind regards,

Huijuan Cao, Ph.D.

Academic Editor

PLOS ONE

Journal Requirements:

"Fan Liu is supported by the Sanming Project of Medicine in Shenzhen (SZSM201612001).Zhuo-xin Yang is supported by National veteran Chinese medicine expert inheritance studio construction project ([2022]75). The funders had no role in study design, data collection and analysis, decision to publish, or preparation of the manuscript."

Please provide an amended statement that declares *all* the funding or sources of support (whether external or internal to your organization) received during this study, as detailed online in our guide for authors at http://journals.plos.org/plosone/s/submit-now.  

Please also include the statement “There was no additional external funding received for this study.” in your updated Funding Statement. 

Additional Editor Comments:

I think the main problem of the manuscript is that the report on some key issues is not detailed, for example, the sample size estimation method, the sampling method and the professional background of the interviewees are not explained in detail. As the author mentioned in the discussion, if the sampling method is not random, or if the interviewee's experience of acupuncture treatment for PPD is not enough, the results of the interview may not be representative, so the conclusion needs to be reconsidered. At the same time, the author's report the results of data analysis is too complicated. It is suggested to give a table to summarize the theme, sub theme and codes extracted from the interview, and then explain and analyze them one by one. Although the author mentioned the report of the study was guided by the COREQ, the checklist should be attached as an appendix and the content of the manuscript should be checked again according to the checklist.

Please report the qualitative study according to the consolidated criteria for reporting qualitative health research checklist (COREQ), which you may find in EQUATOR website (https://www.equator-network.org/) or in below publication:

Tong A, Sainsbury P, Craig J. Consolidated criteria for reporting qualitative research (COREQ): a 32-item checklist for interviews and focus groups. Int J Qual Health Care. 2007; 19: 349–357; https://doi.org/10.1093/intqhc/mzm042

Reviewers' comments:

Reviewer's Responses to Questions

**Comments to the Author**

1. Is the manuscript technically sound, and do the data support the conclusions?

Reviewer #1: Yes

Reviewer #2: Yes

Reviewer #3: Partly

2. Has the statistical analysis been performed appropriately and rigorously? 

Reviewer #1: No

Reviewer #2: N/A

Reviewer #3: N/A

3. Have the authors made all data underlying the findings in their manuscript fully available?

Reviewer #1: Yes

Reviewer #2: Yes

Reviewer #3: No

4. Is the manuscript presented in an intelligible fashion and written in standard English?

Reviewer #1: Yes

Reviewer #2: Yes

Reviewer #3: Yes

5. Review Comments to the Author

Reviewer #1: This thesis is a subsidiary study of the Clinical Registry Study of Acupuncture for PDD. The study used a qualitative analysis to interview 14 clinicians within the registry system. The subject matter was interesting, but there were some methodological and writing issues.

1. All interviews for the study were conducted by independent authors. However, the process of interview outline formation was not described. Were other members of the team involved in relevant discussions? If so, the basic information about these members needs to be included.

2. How were the study participants identified? Were the 16 invited physicians all frontline therapists for this registered study? If not, what sampling principles were followed to select these 16 physicians?

3. The process of coding extraction and formation is very important, but is not described. This reduces the reliability of the study.

4. Figure 1 is unnecessary and it does not provide more information than the subheadings.

5. It is recommended to create an expanded flowchart. The flowchart could start with patient recruitment and go through treatment, data collection, and up to the completion of the clinical protocol and then out of the group. Include details of the questions for each process. This will be more informative.

6. The discussion is too superficial and does not demonstrate the authors' interest in the core issues of the study. Not all issues addressed in the paper need to be discussed equivalently. The author should have a notion of the importance of all issues, and relatively minor issues do not need to occupy equal space. This could have been a reason to reject the paper, but the work itself is valuable. It would be nice to read a more concise discussion.

7. This study is an affiliated study to a registered clinical study. What does the study recommend for the methodology of similar registry studies thereafter? It would be more informative.

Reviewer #2: In the statements from the acupuncture practitioners, there are some minor English errors. Page 18 comment on top of the page ..."They prefer to adopt it because we are confident in our skills." I would suggest adding [acupuncture] after the work "it" so the meaning is clear.

page 20: "Time is the most important factor, as patients need to breastfeed. In fact, there were not

a few patients who came with their children, which must be a great obstacle" In this statement, it is unclear whether the breastfeeding or having children at the appointment is the issue. While this is a quote, some edits are needed to clarify what the issue is.

page 26: "Time is the most important factor, as patients need to breastfeed. In fact, there were not

a few patients who came with their children, which must be a great obstacle.." For this sentence, I believe the "he" needs to be replaced with "she"

Page 27: "Because of the wide variety of acupuncture treatments, multiple acupuncture therapies can be combined to accomplish the goal of synergistic and synergistic enhancement." There must be a typo here. Synergistic does not need to be used twice.

Reviewer #3: It is interesting that authors conducted a qualitative study of practioners’ perspectives on acupuncture treatment for postpartum depression, I think several concerns listed below will need to be addressed to improve paper quality

1. The sample size ( n = 14) was too small to generalize the findings of this study. Please provide a rationale for generalizing your findings

2. Please provide the evidence that participants are specialist in acupuncture treatments for postpartum depression.

3. Please provide the raw data summarizing participant’s answers to each question as a supporting file.

4. Please provide the ethical approval date.

6. PLOS authors have the option to publish the peer review history of their article (what does this mean?). If published, this will include your full peer review and any attached files.

Reviewer #1: No

Reviewer #2: **Yes: **Jennifer E Brett

Reviewer #3: No

---

## [Author Response · Author response to Decision Letter 0]

26 Sep 2022

Replies to the Editor

Comment 1: I think the main problem of the manuscript is that the report on some key issues is not detailed, for example, the sample size estimation method, the sampling method and the professional background of the interviewees are not explained in detail. As the author mentioned in the discussion, if the sampling method is not random, or if the interviewee's experience of acupuncture treatment for PPD is not enough, the results of the interview may not be representative, so the conclusion needs to be reconsidered.

Response: Thank you for pointing out the sampling issue. As far as we are aware, the use of acupuncture to treat PPD in China is still fairly limited; hence, the number of experienced acupuncturists is small. This study is embedded within the multicentre registry trial, allowing us to contact acupuncturists with experience in the treatment of PPD. In the revised manuscript, we have added basic respondent inclusion criteria. We did our best to invite all competent frontline acupuncturists to participate, but information saturation cannot be guaranteed, as explained in the discussion section.

Comment 2: The author's report the results of data analysis is too complicated. It is suggested to give a table to summarize the theme, sub theme and codes extracted from the interview, and then explain and analyze them one by one.

Response: Thank you for your suggestion. We have supplemented the table (Table 2) in the revised manuscript. Page 10-13

Comment 3: Although the author mentioned the report of the study was guided by the COREQ, the checklist should be attached as an appendix and the content of the manuscript should be checked again according to the checklist.

Response: At your request, we checked the content of the manuscript again according to the COREQ checklist, and the revised manuscript has been supplemented with the checklist as an appendix.

Replies to Reviewer #1

Comment 1: All interviews for the study were conducted by independent authors. However, the process of interview outline formation was not described. Were other members of the team involved in relevant discussions? If so, the basic information about these members needs to be included.

Response: Thank you for your query. The revised manuscript supplements the interview outline formation process. The research team (FL, TYZ, YMZ, YYZ, and ZXY) participated in the formulation of the research questions, which are mentioned in the revised manuscript. Page7-8

Comment 2: How were the study participants identified? Were the 16 invited physicians all frontline therapists for this registered study? If not, what sampling principles were followed to select these 16 physicians?

Response: We apologize that we did not clarify the selection criteria for the respondents.

The following revisions were added to the manuscript: Based on a purposive sampling strategy, all frontline acupuncture practitioners in the registry study who met the following criteria were invited to participate in the qualitative interview study: practitioners who were involved in the clinical trial for more than 1 year and those who treated at least 20 patients. Ultimately, 16 acupuncturists from 7 Class-A tertiary hospitals met the requirements: One acupuncturist declined to participate, 1 did not respond, and 14 agreed to participate.

Comment 3: The process of coding extraction and formation is very important, but is not described. This reduces the reliability of the study.

Response: Thank you for your comment. The process of coding extraction and formation is summarized in a table (Table 2) that has been added to the revised manuscript. Page 10-12

Comment 4: Figure 1 is unnecessary and it does not provide more information than the subheadings.

Response: Thank you for your suggestion. We have removed Figure 1 in the revised manuscript.

Comment 5: It is recommended to create an expanded flowchart. The flowchart could start with patient recruitment and go through treatment, data collection, and up to the completion of the clinical protocol and then out of the group. Include details of the questions for each process. This will be more informative.

Response: Thank you very much for your constructive advice. Following your guidance, we have supplemented the flowchart as Figure 1 in the revised manuscript. Page 8

Comment 6: The discussion is too superficial and does not demonstrate the authors' interest in the core issues of the study. Not all issues addressed in the paper need to be discussed equivalently. The author should have a notion of the importance of all issues, and relatively minor issues do not need to occupy equal space. This could have been a reason to reject the paper, but the work itself is valuable. It would be nice to read a more concise discussion.

Response: We appreciate the critical comment. Based on the core issue of the clinical feasibility of acupuncture for PPD, we have reworked the Discussion section to focus on the advantages and drawbacks of acupuncture for PPD as well as suggestions for future improvements. We hope that the updated Discussion section is more concise and concentrated.

Comment 7: This study is an affiliated study to a registered clinical study. What does the study recommend for the methodology of similar registry studies thereafter? It would be more informative.

Response: We apologize that this study did not involve the issue of registry research methodology, and we thank you for your suggestion to help us improve the design of future research protocols.

Replies to Reviewer #2

Thank you very much for pointing out the errors in English usage. The above errors have been corrected in the revised manuscript. In addition, this manuscript has been edited and proofread by AJE for language-related concerns.

Replies to Reviewer #3

Comment 1: The sample size ( n = 14) was too small to generalize the findings of this study. Please provide a rationale for generalizing your findings.

Response: Thank you for pointing out the sample size issue. As far as we are aware, the use of acupuncture to treat PPD in China is still fairly limited; hence, the number of experienced acupuncturists is small. This study is embedded within the multicentre registry trial, allowing us to contact acupuncturists with experience in the treatment of PPD. In the revised manuscript, we have added basic respondent inclusion criteria. We did our best to invite all competent frontline acupuncturists to participate, but information saturation cannot be guaranteed, as explained in the discussion section.

Nevertheless, the findings drawn through the rigorous data analysis process are still helpful in answering the research questions.

Comment 2: Please provide the evidence that participants are specialist in acupuncture treatments for postpartum depression.

Response: Thank you for your comment. All the participants were from the multicentre registry study. According to the research plan, an acupuncture practitioner needed a physician's qualification certificate and licence as well as at least three years of work experience in the acupuncture department. Before the study, the practitioner received training on PPD acupuncture operation standards. In addition, we have supplemented the inclusion criteria in the revised manuscript as follows: practitioners who were involved in the clinical trial for more than 1 year and those who treated at least 20 patients. 

Comment 3: Please provide the raw data summarizing participant’s answers to each question as a supporting file.

Response: Thank you for your comment. In ethically reviewed research protocols, the informed consent forms contain undisclosed respondents' raw data. Data cannot be shared publicly because recordings of the conversations are private and contain potentially identifying and personal information. Data are available from the Ethics Committee of Shenzhen Traditional Chinese Medicine Hospital for researchers who meet the criteria for access to confidential data. The data underlying the results presented in the study are available via the following email address: szszyyll@126.com.

Comment 4: Please provide the ethical approval date.

Response: Thank you for your careful reading. We have added the ethical approval date to the revised manuscript. Page 7

---

## [Decision Letter · Decision Letter 1]

17 Nov 2022

PONE-D-22-18861R1Practitioners’ perspectives on acupuncture treatment for postpartum depression: A qualitative studyPLOS ONE

Dear Dr. Yang,

Thank you for submitting your manuscript to PLOS ONE. After careful consideration, we feel that it has merit but does not fully meet PLOS ONE’s publication criteria as it currently stands. Therefore, we invite you to submit a revised version of the manuscript that addresses the points raised during the review process.

We look forward to receiving your revised manuscript.

Kind regards,

Huijuan Cao, Ph.D.

Academic Editor

PLOS ONE

Reviewers' comments:

Reviewer's Responses to Questions

**Comments to the Author**

1. If the authors have adequately addressed your comments raised in a previous round of review and you feel that this manuscript is now acceptable for publication, you may indicate that here to bypass the “Comments to the Author” section, enter your conflict of interest statement in the “Confidential to Editor” section, and submit your "Accept" recommendation.

Reviewer #2: All comments have been addressed

Reviewer #3: (No Response)

Reviewer #4: (No Response)

Reviewer #5: All comments have been addressed

2. Is the manuscript technically sound, and do the data support the conclusions?

Reviewer #2: Yes

Reviewer #3: Partly

Reviewer #4: Yes

Reviewer #5: No

3. Has the statistical analysis been performed appropriately and rigorously? 

Reviewer #2: Yes

Reviewer #3: I Don't Know

Reviewer #4: I Don't Know

Reviewer #5: N/A

4. Have the authors made all data underlying the findings in their manuscript fully available?

Reviewer #2: Yes

Reviewer #3: No

Reviewer #4: Yes

Reviewer #5: Yes

5. Is the manuscript presented in an intelligible fashion and written in standard English?

Reviewer #2: Yes

Reviewer #3: Yes

Reviewer #4: Yes

Reviewer #5: Yes

6. Review Comments to the Author

Reviewer #2: All reviewer comments and concerns have been addressed in this updated version of the manuscript. The article is acceptable as written.

Reviewer #3: Thank you for your response to my comments. However, I am not satisfied with your response. The quality of this manuscript is low due to small sample size and the ambiguity of the research process. You should include frontline acupuncture practitioners who were not involved in the registry study.

Reviewer #4: GENERAL: The text is not paged for ease of assessment.

METHODS: Under study design, the authors should define what they mean by class A tertiary hospitals. What do the authors mean by postdoc? Under ethical considerations, how were written informed consents obtained from the participants that resided for away and had to be interviewed by telephone?

REFERENCES: This is ok.

Reviewer #5: The manuscript is concerned about the small sample size and results. In addition, the results are not quantified, making objective understanding difficult.

Acupuncture is indicated in the title, but the effect of acupuncture itself is not shown. Because of this, the correct word to include in the title would be “acupoint stimulation”.

In addition, It should be noted why were the acupuncture points used? For example, Historically, what are the benefits of each targeted acupuncture points for this symptoms?

Various techniques are used to stimulate the acupuncture points (e.g. auricular acupuncture, cupping, moxibustion) in the method paragraph. But it will make the outcome mor difficult.

Please provide evidence that the title is true to show the effectiveness of acupuncture in the presence of other stimuli.

Write the method of stimulation details such as needle type, stimulation frequency or intensity, insert depth, and duration of acupuncture care. In addition, consider the differences in the stimuli from previous studies.

7. PLOS authors have the option to publish the peer review history of their article (what does this mean?). If published, this will include your full peer review and any attached files.

Reviewer #2: **Yes: **Jennifer Brett, ND, L.Ac.

Reviewer #3: No

Reviewer #4: No

Reviewer #5: No

---

## [Author Response · Author response to Decision Letter 1]

25 Dec 2022

Reviewers' comments

Reviewer #3: Thank you for your response to my comments. However, I am not satisfied with your response. The quality of this manuscript is low due to small sample size and the ambiguity of the research process. You should include frontline acupuncture practitioners who were not involved in the registry study.

Replies to Reviewer #3

We apologize that our response did not meet your expectations. Please allow us to elaborate further on the sample size. Power calculations determine the sample size (N) necessary to demonstrate effects of a certain magnitude from an intervention in quantitative studies. However, for qualitative interview studies, no similar standards for the assessment of sample size exist. Compared to quantitative research, qualitative research typically employs a small sample size. However, the small sample size does not indicate low quality of the study. Qualitative research is largely exploratory and aims to deepen our understanding of a topic or phenomenon by providing new insights. Research with social constructivist roots, where knowledge is considered partial, intermediate, and dependent on the situated view of the researcher, does not support the idea that qualitative studies ideally should comprise a “total” amount of facts. Some scholars propose the concept of “information power”[1], which suggests that the more information the sample holds that is relevant for the actual study, the lower the number of participants that is needed. In this regard, sample adequacy, data quality, and the variability of relevant events are often more important than the number of participants.

Regarding your suggestion to include frontline acupuncture practitioners who were not involved in the registry study, this approach was inconsistent with the goal of our research. Only acupuncturists with sufficient experience in the treatment of postpartum depression could adequately comment on this subject. Based on the results of the preliminary exploratory research, we will invite more frontline acupuncturists if we conduct consensus research in the future.

Regarding your concern about the ambiguity of the research procedure, we must admit that due to the subjectivity and flexibility of qualitative research, quality control and evaluation are more difficult than they are in quantitative research. We have attempted to adhere to the items of the Consolidated Criteria for Reporting Qualitative Research (COREQ) to ensure that reflexivity, transparency and interactivity were implemented in the design, conduct and reporting stages of the qualitative interviews.

[1] Malterud K, Siersma V D, Guassora A D. Sample size in qualitative interview studies: guided by information power[J]. Qualitative health research, 2016, 26(13): 1753-1760.

Reviewers' comments

Reviewer #4: GENERAL: The text is not paged for ease of assessment.

METHODS: Under study design, the authors should define what they mean by class A tertiary hospitals. What do the authors mean by postdoc? Under ethical considerations, how were written informed consents obtained from the participants that resided for away and had to be interviewed by telephone?

REFERENCES: This is ok.

Replies to Reviewer #4

Comment 1: Under study design, the authors should define what they mean by class A tertiary hospitals.

Response: Thank you for your comment. Class A tertiary hospitals are the highest level of hospitals in mainland China and are classified in accordance with the Measures for Graded Management of Hospitals.

The classification of Chinese hospitals is a 3-tier system including primary, secondary and tertiary hospitals. A tertiary hospital is a comprehensive, referral, general hospital at the city, provincial or national level with a bed capacity exceeding 500. These hospitals are responsible for providing specialist health services, perform a larger role in medical education and scientific research and serve as medical hubs that provide care to multiple regions. A tertiary hospital is similar to a tertiary referral hospital in the West. Based on the level of service provision, size, medical technology, medical equipment, and management and medical quality, these 3 grades are further subdivided into 3 subsidiary levels: A, B and C.

Comment 2: What do the authors mean by postdoc?

Response: Thank you for your query. According to the consolidated criteria for reporting qualitative studies (COREQ) 32-item checklist, researchers should describe personal characteristics such as their credentials, occupation, gender, experience and training to show their reflexivity. This interview study was conducted by the first author, who was a postdoctoral fellow working in Shenzhen Traditional Chinese Medicine Hospital at the time of the study.

Comment 3: Under ethical considerations, how were written informed consents obtained from the participants that resided for away and had to be interviewed by telephone?

Response: Thank you for your query. Written informed consent was collected by post or fax from participants who were interviewed via telephone.

Reviewers' comments

Reviewer #5: The manuscript is concerned about the small sample size and results. In addition, the results are not quantified, making objective understanding difficult.

Acupuncture is indicated in the title, but the effect of acupuncture itself is not shown. Because of this, the correct word to include in the title would be “acupoint stimulation”.

In addition, It should be noted why were the acupuncture points used? For example, Historically, what are the benefits of each targeted acupuncture points for this symptoms?

Various techniques are used to stimulate the acupuncture points (e.g. auricular acupuncture, cupping, moxibustion) in the method paragraph. But it will make the outcome more difficult.

Please provide evidence that the title is true to show the effectiveness of acupuncture in the presence of other stimuli.

Write the method of stimulation details such as needle type, stimulation frequency or intensity, insert depth, and duration of acupuncture care. In addition, consider the differences in the stimuli from previous studies.

Replies to Reviewer #5

Thank you for your valuable comments. As you mentioned, the small sample size and the lack of quantitative results are disadvantages of qualitative research compared to quantitative research. Qualitative research aims to address questions concerned with developing an understanding of the meaning and experience dimensions of humans’ lives and social worlds. The data are analysed by summarizing, categorizing and interpreting, so the results are mainly expressed in words. Interpretation of the results should be considered within the context of qualitative research and the exploratory nature of the study.

Comment 1: Acupuncture is indicated in the title, but the effect of acupuncture itself is not shown. Because of this, the correct word to include in the title would be “acupoint stimulation”. Various techniques are used to stimulate the acupuncture points (e.g. auricular acupuncture, cupping, moxibustion) in the method paragraph. But it will make the outcome more difficult. Please provide evidence that the title is true to show the effectiveness of acupuncture in the presence of other stimuli.

Response: Thank you for your comments. The qualitative research is an adjunct to the clinical registry study of acupuncture for PPD, the objective of which is to assess the efficacy and safety of acupuncture in the treatment of mild to moderate PPD as well as the advantages of various types of acupuncture, including filiform needle acupuncture, electroacupuncture, and auricular acupuncture. Since this was a real-world study, physicians could select different acupuncture treatment plans based on the conditions of individual patients. Additionally, combination therapies, such as cupping and moxibustion, could be used. Patients undergoing different treatment plans naturally created distinct cohorts, allowing for the improvement of treatment plans through statistical analysis.

In this study, most of the patients only received filiform needle acupuncture. With regard to the effect of acupuncture, our team published a Chinese paper[2] based on a mid-term analysis.

[2]Yan B, Yang Z X, Cui L L, et al. Mild and moderate postpartum depression treated with acupuncture of Tiaoren Tongdu: a real world study[J]. Zhongguo Zhen jiu= Chinese Acupuncture & Moxibustion, 2021, 41(8): 877-882.

Comment 2: In addition, It should be noted why were the acupuncture points used? For example, Historically, what are the benefits of each targeted acupuncture points for this symptoms?

Response: Thank you for your query. Tiaoren Tongdu refers to a collection of acupoints selected primarily from the Ren and Du meridians. It was developed by our research team based on classical literature and clinical experience to treat neuropsychiatric disorders. Its efficacy in treating poststroke depression and insomnia has been demonstrated through clinical research.

According to the current literature review on the treatment of depression by acupuncture[3], the core points (GV20, GV29, PC6, HT7, LI4, ST36, SP6, LR3) for the treatment of depression are all included in the Tiaoren Tongdu. Therefore, Tiaoren Tongdu represents the conventional acupuncture therapy strategy for depression.

GV20 and GV29 are located in the Du meridian (Governor Vessel meridian) on the head and have the effect of regulating the mind (Shen) and relieving depression. CV12, CV6 and CV4 are located in the Ren meridian (Conception Vessel meridian), adjacent to the uterus, and can replenish vitality and kidney essence.The Du meridian is dominant to the Yang meridian, while the Ren meridian is dominant to the Yin meridian. Ren and Du meridian acupuncture can balance the Yin and Yang of the human body.

According to traditional Chinese medicine, the heart controls mental activities. PC6 lies on the pericardium meridian, whereas HT7 is on the heart meridian. These acupoints are frequently used to nourish the mind and calm the nerves.

Traditional Chinese medicine suggests that depression is mostly related to the stagnation of liver qi. LR3 and LI 4 are typical combinations of acupoints that can help regulate liver qi. Traditional Chinese medicine also suggests that the spleen and stomach are the foundation of acquired life and the source of qi and blood. SP6 is located in the spleen meridian, which has the ability to generate blood. ST36 is located in the stomach meridian, which helps replenish qi. The combination of two acupoints can replenish qi and blood. Therefore, this combination is suitable for the constitution of postpartum women.

[3]Sun Z Y, Shan X J, Huang X Y, et al. Visual analysis of literature knowledge structure and acupoint matching rules of acupuncture for depression[J]. Zhongguo Zhen jiu= Chinese Acupuncture & Moxibustion, 2021, 41(9): 1049-1054.

Comment 3: Write the method of stimulation details such as needle type, stimulation frequency or intensity, insert depth, and duration of acupuncture care. In addition, consider the differences in the stimuli from previous studies.

Response: Thank you for your comment. The needle type we used was 0.3 mm × 40 mm/0.3 mm × 75 mm. For the GV20 and GV29 acupoints, the needles are inserted horizontally at 0.5-0.8 cun to induce a sensation of soreness (de qi). For the PC6, HT7, LI4, ST36, SP6, and LR3 acupoints, the needles are inserted vertically to a depth of 0.8-1.0 cun. For the CV4, CV6 and CV12 acupoints, the needles are inserted obliquely at 1.0-1.2 cun. Acupuncturists rotate and lift the needles for 30 s to achieve de qi. The treatment period is 8 weeks, and the treatment frequency is three times per week, for a total of 24 acupuncture treatments over 8 weeks.

Conventional filiform needle acupuncture therapy was used in this study, which does not differ from the stimuli in previous studies.

---

## [Decision Letter · Decision Letter 2]

6 Feb 2023

PONE-D-22-18861R2Practitioners’ perspectives on acupuncture treatment for postpartum depression: A qualitative studyPLOS ONE

Dear Dr. Yang,

Thank you for submitting your manuscript to PLOS ONE. After careful consideration, we feel that it has merit but does not fully meet PLOS ONE’s publication criteria as it currently stands. Therefore, we invite you to submit a revised version of the manuscript that addresses the points raised during the review process.

We look forward to receiving your revised manuscript.

Kind regards,

Huijuan Cao, Ph.D.

Academic Editor

PLOS ONE

Journal Requirements:

Additional Editor Comments:

The sample size of qualitative research generally adopts the principle of information saturation. The author also fully mentions the uncertainty of the results caused by the sampling method in the limitations sesstion. I suggest the authors to add whether (all or most of) the respondents are from the same hospital. If so, please also explain the general information of acupuncture treatment for postpartum depression in that hospital.

Reviewers' comments:

Reviewer's Responses to Questions

**Comments to the Author**

1. If the authors have adequately addressed your comments raised in a previous round of review and you feel that this manuscript is now acceptable for publication, you may indicate that here to bypass the “Comments to the Author” section, enter your conflict of interest statement in the “Confidential to Editor” section, and submit your "Accept" recommendation.

Reviewer #3: (No Response)

Reviewer #5: All comments have been addressed

2. Is the manuscript technically sound, and do the data support the conclusions?

Reviewer #3: No

Reviewer #5: Partly

3. Has the statistical analysis been performed appropriately and rigorously? 

Reviewer #3: I Don't Know

Reviewer #5: I Don't Know

4. Have the authors made all data underlying the findings in their manuscript fully available?

Reviewer #3: No

Reviewer #5: No

5. Is the manuscript presented in an intelligible fashion and written in standard English?

Reviewer #3: Yes

Reviewer #5: Yes

6. Review Comments to the Author

Reviewer #3: Thank you for your response to my comments. However, I am not satisfied with your response. I still think the quality of this manuscript is low due to small sample size and the ambiguity of the research process

Reviewer #5: They answered my questions. The authors understood that the study was not objectively evaluable but explained it carefully. Oriental medical stimulation also seems to be suitable for the purpose.

7. PLOS authors have the option to publish the peer review history of their article (what does this mean?). If published, this will include your full peer review and any attached files.

Reviewer #3: No

Reviewer #5: No

---

## [Author Response · Author response to Decision Letter 2]

8 Feb 2023

Journal Requirements:

Additional Editor Comments:

The sample size of qualitative research generally adopts the principle of information saturation. The author also fully mentions the uncertainty of the results caused by the sampling method in the limitations section. I suggest the authors to add whether (all or most of) the respondents are from the same hospital. If so, please also explain the general information of acupuncture treatment for postpartum depression in that hospital.

Replies to Editor:

Thank you for your suggestion. In the method section (Participants and setting /p5), we mentioned that the respondents were from 7 hospitals, and in the results section (Acupuncture treatment methods/p18), we introduced the general information of acupuncture treatment for PPD.

The references have been reviewed to ensure accuracy.

---

## [Editor Report · Decision Letter 3]

21 Feb 2023

Practitioners’ perspectives on acupuncture treatment for postpartum depression: A qualitative study

PONE-D-22-18861R3

Dear Dr. Yang,

We’re pleased to inform you that your manuscript has been judged scientifically suitable for publication and will be formally accepted for publication once it meets all outstanding technical requirements.

Kind regards,

Huijuan Cao, Ph.D.

Academic Editor

PLOS ONE
---

## [Editor Report · Acceptance letter]

24 Feb 2023

PONE-D-22-18861R3 

Practitioners’ perspectives on acupuncture treatment for postpartum depression: A qualitative study 

Dear Dr. Yang:

I'm pleased to inform you that your manuscript has been deemed suitable for publication in PLOS ONE. Congratulations! Your manuscript is now with our production department. 

Kind regards, 

on behalf of

Dr. Huijuan Cao 

Academic Editor

PLOS ONE